# Immune marker reductions in black and white Americans following sleeve gastrectomy in the short-term phase of surgical weight loss

Charles L. Phillips[1,2], Tran T. Le[3], Seth T. Lirette[3], Bradley A. Welch[2], Sarah C. Glover[4], Adam Dungey[5], Kenneth D. Vick[5], Bernadette E. Grayson[2]*

1 Program in Pathology, University of Mississippi Medical Center, Jackson, Mississippi, United States of America, 2 Department of Neurobiology and Anatomical Sciences, University of Mississippi Medical Center, Jackson, Mississippi, United States of America, 3 Department of Data Science, University of Mississippi Medical Center, Jackson, Mississippi, United States of America, 4 Department of Internal Medicine, University of Mississippi Medical Center, Jackson, Mississippi, United States of America, 5 Department of Surgery, University of Mississippi Medical Center, Jackson, Mississippi, United States of America

* bgrayson@umc.edu

## Abstract

### Background

Surgical weight loss procedures like vertical sleeve gastrectomy (SG) are sufficient in resolving obesity comorbidities and are touted to reduce the burden of pro-inflammatory cytokines and augment the release of anti-inflammatory cytokines. Recent reports suggest a reduced improvement in weight resolution after SG in Black Americans (BA) versus White Americans (WA). The goal of this study was to determine if differences in immunoglobulin levels and general markers of inflammation after SG in Black Americans (BA) and White Americans (WA) may contribute to this differential resolution.

### Methods

Personal information, anthropometric data, and plasma samples were collected from 58 participants (24 BA and 34 WA) before and 6 weeks after SG for the measurement of immunoglobulin A (IgA), IgG, IgM, C-reactive protein (CRP), and transforming growth factor (TGFβ). Logistic regression analysis was used to determine the relationship of measures of body size and weight and inflammatory markers.

### Results

Both IgG and CRP were significantly elevated in BA in comparison to WA prior to weight loss. Collectively, IgG, TGFβ, and CRP were all significantly reduced at six weeks following SG. CRP levels in BA were reduced to a similar extent as WA, but IgG levels were more dramatically reduced in BA than WA despite the overall higher starting concentration. No change was observed in IgA and IgM.

**Data Availability Statement:** All relevant data are within the manuscript. Metadata are found at Figshare after embargo https://doi.org/10.6084/m9.figshare.22354738.v1.

**Funding:** The Mississippi Center of Excellence in Perinatal Research is supported by National Institute of General Medical Sciences P20GM121334. The Mississippi Center for Clinical and Translational Research is supported by National Institute of General Medical Sciences, 1U54GM115428-06. BEG and SL received core support from both of these center grants. The content is solely the responsibility of the authors and does not necessarily represent the official views of the National Institutes of Health. The funders had no role in study design, data collection and analysis, decision to publish, or preparation of the manuscript.

**Competing interests:** The authors have declared that no competing interests exist.

## Conclusions

These data suggest that SG improves markers of immune function in both BA and WA. More diverse markers of immune health should be studied in future work.

## Introduction

Obesity and its associated comorbidities continue to present a strain on health care around the world. Obesity is a disease of excess adiposity accompanied by chronic low-grade inflammation. The expanding size and number of adipocytes stress the surrounding tissues causing active secretion of cytokines from the adipose and endothelial cells and infiltration of immune cells into the surrounding tissues [1]. Over time, inflammatory markers are elevated enough to be reliably measurable in circulation. The elevated inflammation further drives both systemic and local immune cell activation and cytokine secretion resulting in diverse negative impacts on various organ systems including immuno-hematologic parameters [2].

Black Americans (BA) carry a larger burden of obesity-related diseases than White Americans (WA). BA women have the greatest prevalence of obesity in the United States [3]. Among BA women, 80% are overweight or obese in comparison to only 62.4% of WA women [4]. 34.6% of BA women are Class II and III Obese compared to 17.6% of WA women [4]. Further, among BA men and women, the age-adjusted prevalence of hypertension is 39.6% and 43.1%, respectively [5]. In WA men and women, the prevalence is 31.4% and 28.7%, respectively [5]. Racial disparities also exist in the incidence of T2DM; 18.7% of all BA >20 years of age have T2DM compared to 7.1% of WA [6]. As a whole, BA suffer from a higher rate of metabolic-related diseases in comparison to WA.

In the U.S., BA also carry a higher burden of inflammation than WA. In a study of racial and ethnic disparities in children, the risk of inflammation is higher in BA children in comparison to WA children [7]. Low parental education and elevated child BMI were partial mediators of this higher association [7]. In a cross-sectional investigation of ethnicity and blood levels of inflammatory markers in non-smoking, church-goers, BA had higher CRP and IL-6 than WA [8]. Further, in a study comparing inflammatory markers and breast cancer risk factors, BA women had higher levels of inflammatory cytokines IL-6 and interferon γ [9]. In a study of the effect of sleep and loss of immunity, BA participants had higher levels of Il-6 and IL-10 than WA [10]. Finally, BA patients higher IgG levels in a variety of studies, regardless of context [11, 12]. Taken together, significant evidence in a variety of different sectors of population suggests that BA have higher levels of inflammatory mediators than WA.

Surgical weight loss procedures are sufficient in the resolution of obesity comorbidities such as excess weight, diabetes, and hypertension [13]. In a recent study of over 14,000 patients of which half were BA, they reported that BA have reduced improvements to body weight loss and percentage of total weight loss in comparison to WA [14]. Further, BA patients experienced significantly reduced remission of hypertension but there was no difference between WA and BA patients in remission of insulin-dependent diabetes [14]. Higher overall rates of complications and health care resource utilization in BA have also been reported within 30 days of surgery in comparison to controls [14]. Bariatric surgeries are touted to reduce the burden of pro-inflammatory cytokines and augment release of anti-inflammatory cytokines [15, 16]. Limited reports exist on immunoglobulins (Ig) status after bariatric procedures. In a small cohort of patients in China obtaining SG, IgA, IgG and IgM) are elevated after surgical weight loss at two and six months post-surgery in comparison to pre-surgical levels [17]. In another small cohort of patients in Korea, IgG levels were reduced six months following laparoscopic

SG [18]. Further c-reactive protein (CRP) levels are generally reduced at 6-months post-surgery in comparison to peri-operative levels [18, 19] Other inflammatory cytokines such as IL-6 and IL-8 were elevated in the weeks following surgical weight loss. IL-8 continued to be elevated three months after surgery with no change to IL-6 and TGF- β [20]. Moreover, bariatric surgery alters immune cell function and proliferation [21–23]. Nevertheless, the impact of surgery on inflammation in comparing BA and WA surgical recipients has not been reported.

Given that the burden of obesity-related disease is substantial in BA, and inflammation is higher in BA independent obesity and its related comorbidities, understanding the effect of surgical weight loss outcomes in BA is important. The focus of this study was to determine if bariatric surgery ameliorates obesity-related inflammation in BA to a similar extent as seen in WA subjects. Vertical sleeve gastrectomy (SG) is the most common surgery currently performed in the U.S. Here we report some key differences in the early changes of inflammatory markers after SG in a patient population of Mississippi, a state with one of the highest burdens of obesity-related comorbidities. Using anthropometric data and plasma samples from patients before and six weeks following SG, we investigated specific inflammatory markers IgG, IgM, IgA, TGFβ, and CRP.

## Materials and methods

### Assurances

Institutional Review Board (IRB) approval was obtained from the University of Mississippi Medical Center (UMMC), Jackson, MS for the Predictors of Weight Loss (POWL) Study, (IRB# 2014–0047). All procedures were performed in accordance per the 1964 Declaration of Helsinki ethical standards. Written informed consent was obtained from each participant before formally entering the approved study.

### Study design

This is a non-randomized prospective study of obese patients receiving elective SG surgery through the Weight Management Clinic, UMMC were consented. The inclusion criteria for this study comprised of men and women between the ages of 21–65 years, BMI $\geq$35 kg/m$^2$, and undergoing first-time bariatric procedures between June and December 2016. The Weight Management Clinic manages weight remission first through non-surgical methods (diet, exercise and pharmacotherapy) of which a percentage of patients transition to surgical weight loss procedures. In addition, referrals are made from other non-surgical weight management clinics for bariatric surgery at UMMC. Patients were accepted into the study regardless of their previous weight loss attempts. Exclusion criteria for participation in this study were as follows: 1) Individuals with major organ system failure like: cirrhosis, hepatic insufficiency, portal hypertension, severe renal insufficiency or on dialysis, severe arterial insufficiency, dementia, or the inability to give informed consent. 2) Individuals who are pregnant or lactating. 3) Individuals with prior surgical weight loss procedure.

### REDCap electronic record

Study data were collected and managed using REDCap (Research Electronic Data Capture) tools [24] hosted at the University of Mississippi Medical Center. REDCap is a secure, web-based application designed to support data capture for research studies, providing 1) an intuitive interface for validated data entry; 2) audit trails for tracking data manipulation and export procedures; 3) automated export procedures for seamless data downloads to common statistical packages; and 4) procedures for importing data from external sources.

## Plasma analytes

Blood was collected in EDTA coated tubes, processed for plasma and stored at -80 °C until further use. The following kits were used: C-reactive protein (#80955, CrystalChem, Elk Grove Village, IL), immunoglobulin A (#88–50600, Thermofisher, Waltham, MA), immunoglobulin G (#88–50550, Thermofisher, Waltham, MA), immunoglobulin M (#88–50620, Thermofisher, Waltham, MA), and tumor growth factor beta (#RAB0460, SigmaAldrich, St. Louis, MO). All assays were performed according to the manufacturers' specifications.

## Statistical analysis

Descriptive statistics were compiled where appropriate. For comparing post vs. pre-operative characteristics of participants, paired student's T tests and two-way analysis of variance with repeated measures for time were used followed by Tukey's post hoc test for variables of race and time with results are given as means ± SEM (Tables 1 and 2, Fig 1). For regression modeling, all models were adjusted for age, sex, race, hypertension, hyperlipidemia, sleep apnea, and diabetes. Models for pre-op immune markers outcomes consisted of generalized linear models with gamma families to account for right skewness of the markers and identity link to facilitate ease of interpretation (Table 3). Weight and BMI change outcomes were modeled using ordinary least squares on the difference in weight post-op vs. pre-op (Table 4). Similar models were used to model immune marker change outcomes, additionally adjusting for BMI change, systolic change, and diastolic change (Table 5). Multivariable fractional polynomials were used as a first step in modeling to account for potential nonlinearities. All linearity assumptions were found to be valid. Results were considered statistically significant when $p < 0.05$. All statistical analyses were performed with GraphPad Prism v8.1.2 (GraphPad Software, San Diego, California) and Stata v16.1 (StataCorp, College Station, Texas).

## Results

### Baseline characteristics of participants

The total number of participants for the current study was 58 individuals of which 24 self-identified as BA and 34 self-identified as WA. Of the BA participants, all 24 were female, comprising 41.3% of the total; we were unable to enroll any BA males. Of the 34 WA participants, 28 or 48.2% were females and 6 participants or 10.3% of the total were WA males. There was no difference in the average age of female BA and WA participants (average age ± SEM, 45.5±2.0

**Table 1. Baseline characteristics and comorbidities of study participants.** Data presented as percentages. Comparisons made using Chi-squared test.

| Participant Characteristics | BA | % | WA | % | Statistics |
|---|---|---|---|---|---|
| Participants/Total (%) | 24/58 | *41.3%* | 34/58 | *58.6%* | - |
| Female/Total (%) | 24/58 | *41.3%* | 28/58 | *48.2%* | - |
| Male/Total (%) | 0/58 | *0%* | 6/58 | *10.3%* | - |
| Average age (years) females | 45.5±2.0 | | 46.9±2.0 | | - |
| Average age (years) males | N/A | | 43.8±2.7 | | - |
| Pre-existing Hypertension (Y/Total) | 22/24 | *91.6%* | 15/34 | *44.1%* | *$chi^2$ = 13.77; **p<0.001*** |
| Pre-existing Hyperlipidemia (Y/Total) | 4/24 | *16.7%* | 13/34 | *38.2%* | *$chi^2$ = 3.16; P = 0.07* |
| Pre-existing Diabetes (Y/Total) | 9/24 | *37.5%* | 7/34 | *20.5%* | *$chi^2$ = 2.01; P = 0.16* |
| Pre-existing Sleep Apnea (Y/Total) | 11/24 | *45.8%* | 13/34 | *38.2%* | *$chi^2$ = 0.33; P = 0.56* |

Y denotes "yes" for the presence of the comorbidity.

**Table 2. Pre-operative and post-operative characteristics of participants.** Data presented as mean ± SEM. Pre-operative values between BA and WA compared using student T test (Column A vs. B). Two-way Anova was used to compare variables of race and time.

| Measurement | BA | WA | *STATISTIC* | BA | WA | *STATISTIC* | *STATISTIC* |
|---|---|---|---|---|---|---|---|
| | Pre-op (A) | Pre-op (B) | *Student's T* | 6 wks post-op (C) | 6 wks post-op (D) | *Student's T* | *Two-way ANOVA* |
| | Mean ± SEM | Mean ± SEM | *A vs B* | Mean ± SEM | Mean ± SEM | *C vs D* | *time = AB vs CD; race = AC vs. BD* |
| | n = 24 | n = 34 | | n = 24 | n = 34 | | |
| Body weight (kg) | 127.6 ± 4.09 | 123.8 ± 4.02 | *p = 0.5206* | 113.4 ± 3.73 | 108.4 ± 3.52 | *p = 0.3452* | *p(time)<0.0001; p(race) = 0.4292* |
| Body Mass Index (BMI) | 47.5 ± 1.27 | 44.7 ± 1.03 | *p = 0.0934* | 42.9 ± 1.5 | 39.5 ± 1.09 | *p = 0.0349* | *p(time)<0.0001; p(race) = 0.0659* |
| Waist Circumference (cm) | 124.5 ± 2.87 | 128 ± 2.73 | *p = 0.4047* | 115.2 ± 2.83 | 123.7 ± 2.52 | *p = 0.533* | *p(time)<0.0001; p(race) = 0.1826* |
| Hip Circumference (cm) | 134.8 ± 2.78 | 128.2 ± 1.99 | *p = 0.0497* | 112.9 ± 2.32 | 122.5 ± 1.87 | *p = 0.6951* | *p(time)<0.0001; p(race) = 0.8606* |
| Systolic BP (mm Hg) | 143.3 ± 4.31 | 138.6 ± 1.91 | *p = 0.2744* | 134.1 ± 2.65 | 127.7 ± 3.03 | *p = 0.1396* | *p(time)<0.0001; p(race) = 0.1230* |
| Diastolic BP (mm Hg) | 85.4 ± 1.97 | 81.6 ± 2.06 | *p = 0.2051* | 84.8 ± 1.55 | 77.4 ± 2.69 | *p = 0.0374* | *p(time) = 0.1850; p(race)<0.05* |
| Pulse (bpm) | 83.6 ± 3.42 | 79.1 ± 2.05 | *p = 0.2386* | 80.0 ± 2.81 | 74.4 ± 1.99 | *p = 0.0969* | *p(time)<0.05; p(race) = 0.0661* |
| Temperature (⬚ C) | 97.7 ± 0.17 | 97.5 ± 0.14 | *p = 0.3567* | 97.6 ± 0.13 | 96.8 ± 0.94 | *p = 0.4558* | *p(time) = 0.5813; p(race) = 0.278* |
| ALT (IU/L) | 18 ± 1.69 | 36.5 ± 3.82 | *p = 0.0005* | N/A | N/A | | *N/A* |
| AST (IU/L) | 17.7 ± 1.15 | 29.6 ± 3.05 | *p = 0.0033* | N/A | N/A | | *N/A* |
| Billirubin (mg/dL) | 0.4 ± 0.06 | 0.5 ± 0.04 | *p = 0.3286* | N/A | N/A | | *N/A* |
| Creatinine (mg/dL) | 0.8 ± 0.03 | 0.8 ± 0.02 | *p = 0.3576* | N/A | N/A | | *N/A* |
| Glucose (mg/dL) | 104.9 ± 5.11 | 113.6 ± 5.88 | *p = 0.2996* | N/A | N/A | | *N/A* |
| WBC (x10 /ul) | 8 ± 0.54 | 7.3 ± 0.3 | *p = 0.2053* | N/A | N/A | | *N/A* |
| Hematocrit % | 38.7 ± 0.77 | 41.9 ± 0.63 | *p = 0.0026* | N/A | N/A | | *N/A* |
| Platelets (x10 /ul) | 312.5 ± 16.28 | 266 ± 12.13 | *p = 0.0234* | N/A | N/A | | *N/A* |

Boldface designates statistical significance 0.05. BP (blood pressure); ALT (alanine transaminase); AST (aspartate aminotransferase); WBC (white blood cells).

years and 46.9±2.0 years respectively) or WA males (average age of 43.8±2.7 years). The number of persons that were positive for pre-existing Hypertension, Hyperlipidemia, Diabetes and Sleep Apnea were included for each category and numbers of persons with the comorbidity were labeled as "yes (Y). BA participants had significantly higher levels of pre-existing hypertension in comparison to WA patients (91.6% vs.44.1%), *p*<0.001, (Table 1). No difference was measured in the instances of pre-existing hyperlipidemia, diabetes, or obstructive sleep apnea between BA and WA (Table 1).

**Peri-operative and post-operative characteristics of BA and WA patients.** There were no differences in pre-operative body weight between BA and WA participants (Table 2). Six weeks following SG, there was a significant loss in body weight, *p(time)<0.0001*, but no main effect of race (Table 2). Similarly, there was no significant differences in the BMI of the BA and WA participants (Table 2). SG surgery resulted in significantly reduced BMI, *p(time)<0.0001*, with no main effect of race (Table 2). When percent excess body weight (%EWL) is calculated, based on ideal weight, WA participants (Mean ±SEM) (20.55 ± 1.327) had a greater amount of excess bodyweight loss in comparison to BA participants (24.15 ± 0.86), *p<0.05*. Percent weight loss (%WL) was not significant (BA 11.24 ± 0.56 vs. WA 12.44 ± 0.34), nor was the change in BMI (BA 5.32 ± 0.27 vs. WA 5.547 ± 0.20) or percent excess BMI loss (%EBMIL) (BA 25.2 ± 2.07 vs. WA 29.53 ± 1.15).

Similarly, waist circumference was not substantively different in BA and WA participants before SG (Table 2), and though waist measurement diminished with time after surgery, *p(time)<0.0001*, there were no racial differences (Table 2). Hip circumference, on the other hand, was greater in BA participants than WA participants at the peri-operative period, *p<0.05*, and was significantly reduced with SG surgery *p(time)<0.0001* with no differences by

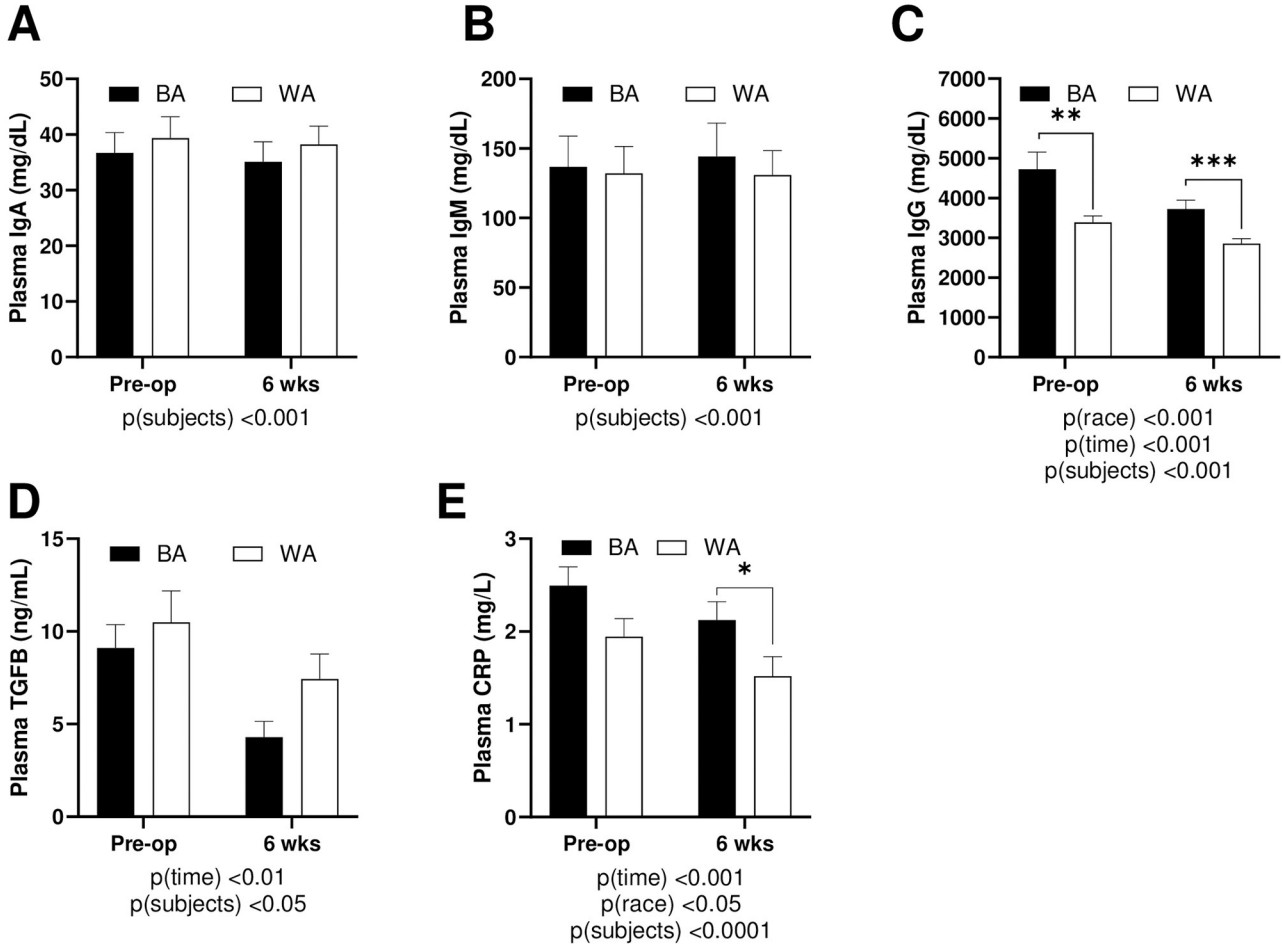

**Fig 1. Immunoglobulins and immune markers pre- and post-SG.** (**A**) Plasma IgA (**B**) Plasma IgG (**C**) IgM (**D**) CRP (**E**) TGFβ. Data presented as mean ± SEM. Two-way Anova was used to compare variables of race and time. IgA (immunoglobulin A); IgG (immunoglobulin G); IgM (immunoglobulin M); CRP (c-reactive protein); TGFB (transforming growth factor beta).

race at six weeks post-surgery (Table 2). There were no starting differences in systolic or diastolic blood pressure prior to SG (Table 2). Systolic blood pressure was uniformly improved after six weeks' time in both BA and WA. However, a significant racial difference in diastolic blood pressure at 6 weeks post-surgery was measured with BA having less of a reduction than WA, p<0.05 (Table 2). There were no differences in resting pulse pre-operatively, and both groups had reductions in pulse rate as an effect of time, *p(time)<0.05* (Table 2) with no specific effect of race. Temperature was similar in BA and WA patients pre-operatively and remained stable after surgery (Table 2).

Blood panels were completed pre-operatively only. Liver enzymes, ALT and AST, were significantly higher in WA participants compared to BA (Table 2). There was no difference observed in bilirubin, *p<0.001* (Table 2). Creatinine and glucose were not different between BA and WA (Table 2). There was no difference in WBC count, though hematocrit percentage was significantly higher in WA, *p<0.01*, and platelets were significantly higher in BA, *p<0.05* (Table 2).

We next developed statistical models to understand early weight change dynamics. Overall, body weight, waist circumference and BMI changes did not correlate with race or the presence

**Table 3. Modelling results for weight change and hip circumference.** Displayed are beta coefficients, p-values, and 95% confidence intervals. Change was defined as post-op minus pre-op, ergo a positive coefficient indicates, on average, less weight loss. Coefficient for age corresponds to one-year increase.

**Models for Weight Change and Hip Circumference (HC)**

|  | Weight Δ (kg) | HC Δ (cm) |
|---|---|---|
| Age | 0.10 **p = 0.045** | -0.04 p = 0.784 |
|  | (0.00, 0.20) | (-0.33, 0.25) |
| Black | -0.09 p = 0.935 | -8.77 **p = 0.004** |
|  | (-2.24, 2.07) | (-14.6, -2.92) |
| Hypertension | 0.67 p = 0.543 | -1.02 p = 0.733 |
|  | (-1.54, 2.88) | (-7.00, 4.96) |
| Hyperlipidemia | 0.75 p = 0.479 | -4.34 p = 0.133 |
|  | (-1.37, 2.88) | (-10.0, 1.37) |
| Sleep Apnea | -1.28 p = 0.223 | 3.93 p = 0.203 |
|  | (-3.36, .80) | (-2.20, 10.1) |
| Diabetes Mellitus | 0.18 p = 0.869 | 3.31 p = 0.278 |
|  | (-2.03, 2.40) | (-2.77, 9.40) |

DM (diabetes mellitus).

of the comorbidities we reported, such as hypertension, hyperlipidemia, sleep apnea, or diabetes mellitus (Table 3). On the other hand, greater reduction in weight were correlated with age, such that younger participants lost weight at an average of 0.10 kg more per year younger and 0.04 points of BMI per year younger than older participants, $p < 0.05$ (Table 3). Further, BA participants had a reduced change in hip circumference (-8.77cm) than WA participants (Table 3).

## Effect of obesity and SG on circulating immune markers

Pre-operative and six weeks post-operative, IgA and IgM levels were not different by race (Fig 1A and 1B). Furthermore, IgA and IgM did not change as a function of surgery (Fig 1A and 1B). Alternately, BA patients had higher levels of IgG than WA patients before SG, *p<0.01* (Fig 1C). All patients experienced reductions in IgG after surgery, *p(time)<0.001*, but plasma IgG varied significantly by race *p*(race)<0.001 with BA having higher levels of IgG than WA at both time points, *p(time)<0.001* as well as at pre-op, *p<0.01* and 6 weeks, *p<0.01* (Fig 1C). TGF-β levels were reduced significantly as a result of SG surgery, but there were no differences by race, before or after surgery, (Fig 1D). Finally, plasma CRP levels tended to be higher in BA patients prior to surgery (Fig 1E). Though both groups realized reductions in CRP *p(time)< 0.001*, BA patients had statistically significantly higher levels of CRP than WA patients at the six-week time point, *p<0.05* (Fig 1E).

We next determined the relationships of these inflammatory markers with the various characteristics of the BA and WA participants first addressing the pre-operative state. Age did not vary with the pre-operative inflammatory markers CRP, IgA, IgG, I, or TGFβ (Table 4). But age did vary the levels of IgM such that younger patients had 3.99 mg/dL reductions per year younger. With respect to IgG, BA participants had on average 1485.5 mg/dL higher IgG levels than WA participants pre-operatively, *p<0.001* (Table 4). Hypertension or sleep apnea status did not vary inflammatory markers tested pre-operatively (Table 4). Hyperlipidemia was associated with 1.49 mg/dL lower CRP, *p<0.001* (Table 4). Pre-existing diabetes was associated with a 17.08 mg/dL reduction on average of IgA in comparison to non-diabetics, *p<0.05*, and

**Table 4. Modelling results of pre-operative immune markers.** Displayed are beta coefficients, p-values, and 95% confidence intervals. Coefficients for age, BMI, SBP, and DBP correspond to one-unit increases.

**Models for PreOp Immune Markers**

|  | IgA | CRP | IgG | IgM | TGFB |
|---|---|---|---|---|---|
| Age | 0.19 p = 0.532 | -0.01 p = 0.435 | 21.47 p = 0.285 | -3.99 p = 0.026 | -0.16 p = 0.158 |
|  | (-0.41,0.80) | (-0.05,0.02) | (-17.91,60.84) | (-7.51,-0.47) | (-0.37,0.06) |
| Black | 3.79 p = 0.586 | -0.60 p = 0.117 | 1485.51 p<0.001 | -43.31 p = 0.267 | -1.34 p = 0.493 |
|  | (-9.86,17.45) | (-1.34,0.15) | (662.46,2308.56) | (-119.72,33.10) | (-5.17,2.49) |
| Hypertension | 6.20 p = 0.467 | 0.20 p = 0.541 | -315.08 p = 0.450 | 18.75 p = 0.599 | 2.12 p = 0.336 |
|  | (-10.48,22.88) | (-0.44,0.85) | (-1132.22,502.05) | (-51.16,88.66) | (-2.20,6.43) |
| Hyperlipidemia | 5.90 p = 0.451 | -1.49 p<0.001 | 501.56 p = 0.210 | -20.23 p = 0.572 | 4.93 p = 0.047 |
|  | (-9.44,21.24) | (-2.05,-0.94) | (-281.78,1284.91) | (-90.31,49.85) | (0.06,9.81) |
| Sleep Apnea | 7.10 p = 0.381 | 0.62 p = 0.060 | -265.17 p = 0.522 | 36.67 p = 0.312 | -1.33 p = 0.523 |
|  | (-8.79,22.99) | (-0.03,1.27) | (-1077.18,546.84) | (-34.42,107.77) | (-5.40,2.75) |
| Diabetes Mellitus | -17.08 p = 0.048 | 0.48 p = 0.082 | -478.66 p = 0.271 | -43.53 p = 0.245 | 3.24 p = 0.241 |
|  | (-34.02,-0.14) | (-0.06,1.02) | (-1331.29,373.96) | (-116.88,29.81) | (-2.17,8.64) |
| PreOp HC | -0.35 p = 0.135 | 0.04 p = 0.001 | 29.49 p = 0.048 | 0.39 p = 0.704 | 0.10 p = 0.155 |
|  | (-0.81,0.11) | (0.02,0.07) | (0.29,58.69) | (-1.62,2.40) | (-0.04,0.25) |
| PreOp SBP | -0.35 p = 0.152 | 0.01 p = 0.463 | 5.82 p = 0.738 | -0.01 p = 0.989 | 0.12 p = 0.123 |
|  | (-0.83,0.13) | (-0.01,0.02) | (-28.20,39.83) | (-1.93,1.90) | (-0.03,0.27) |
| PreOp DBP | 0.15 p = 0.697 | -0.04 p = 0.009 | 7.59 p = 0.684 | 0.05 p = 0.975 | -0.14 p = 0.188 |
|  | (-0.61,0.92) | (-0.06,-0.01) | (-28.92,44.09) | (-2.79,2.88) | (-0.34,0.07) |

Hi SBP (systolic blood pressure); DBP (diastolic blood pressure); CRP (c-reactive protein); IgA (immunoglobulin A); IgG (immunoglobulin G); IgM (immunoglobulin M); TGFB (transforming growth factor beta).

(Table 4). Hip circumference was associated with 0.04 mg/dL increased CRP, p = 0.001, and 29.49 mg/dL IgG levels, *p<0.05* (Table 4). However, higher pre-operative diastolic blood pressure was associated with a significant, albeit subtle, reduction in circulating pre-operative CRP levels by 0.04 mg/L, *p<0.05* (Table 4).

We performed similar modelling for the change (Δ) in the immune markers and various aspects of the participant cohort. Age did not impact the change of any of the cytokines during the first six weeks post-operatively, except for TGFβ, such that for every year older, there was a 0.37 mg/dL increase in TGFβ, p<0.05 (Table 5). Furthermore, BA had greater reductions in IgG, with an average reduction of 985.18 mg/dL as a result of SG in comparison to WA (Table 5). In participants who had hypertension pre-operatively, SG promoted a greater change, on average, of 6.77 mg/dL in TGFβ levels than those who did not have hypertension, *p<0.05* (Table 5). The presence of pre-operative hyperlipidemia, sleep apnea and diabetes mellitus status did not contribute to changes in any of the immune markers measured (Table 5). Change over the 6 weeks following SG in weight, systolic blood pressure, and diastolic blood pressure did not correlate significantly with any of the shifts in the markers of inflammation tested (Table 5). BA had greater reductions in IgG, with an average reduction of 1108 mg/dL as a result of SG in comparison to WA (Table 5). However, this result could be impacted by the finding that BA had demonstrably higher average baseline IgG values (see Table 3) thus making a larger decline possible, compared to WA.

The presence of hypertension or sleep apnea status did not contribute to changes in any of the immune markers measured (Table 5). However, in participants who had diabetes mellitus pre-operatively, SG promoted a greater change, on average, of 27.98 mg/dL in IgM levels than

**Table 5. Modelling results for immune marker change.** Displayed are beta coefficients, p-values, and 95% confidence intervals. Change was defined as post-op minus pre-op, ergo a positive coefficient indicates, on average, less reduction. Coefficients for age, WC, SBP, and DBP correspond to one-unit increases.

| Models for Immune Marker Change | | | | | |
|---|---|---|---|---|---|
| | **Δ IgA** | **Δ IgG** | **ΔIgM** | **Δ CRP** | **ΔTGFB** |
| Age | 0.15 p = 0.577 | 23.04 p = 0.356 | 0.08 p = 0.896 | -0.02 p = 0.094 | 0.37 p = 0.037 |
| | (-0.39,0.70) | (-26.82,72.91) | (-1.13,1.29) | (-0.05,0.00) | (0.02,0.72) |
| Black | -0.70 p = 0.901 | -985.18 p = 0.064 | 0.31 p = 0.981 | 0.29 p = 0.314 | 2.18 p = 0.548 |
| | (-12.08,10.67) | (-2028.39,58.04) | (-25.02,25.64) | (-0.28,0.86) | (-5.10,9.47) |
| Hypertension | -1.70 p = 0.747 | 804.20 p = 0.101 | -3.90 p = 0.739 | -0.25 p = 0.351 | -6.77 p = 0.050 |
| | (-12.25,8.85) | (-163.19,1771.59) | (-27.39,19.59) | (-0.78,0.28) | (-13.52,-0.01) |
| Hyperlipidemia | -5.37 p = 0.317 | -669.23 p = 0.176 | -19.10 p = 0.113 | 0.32 p = 0.230 | -3.29 p = 0.338 |
| | (-16.08,5.34) | (-1651.18,312.72) | (-42.94,4.74) | (-0.21,0.86) | (-10.14,3.57) |
| Sleep Apnea | 0.02 p = 0.998 | -465.08 p = 0.365 | 14.73 p = 0.239 | -0.15 p = 0.604 | -1.05 p = 0.768 |
| | (-11.17,11.21) | (-1491.14,560.97) | (-10.18,39.64) | (-0.71,0.42) | (-8.22,6.11) |
| Diabetes Mellitus | 0.39 p = 0.942 | 789.38 p = 0.118 | 27.98 p = 0.025 | 0.06 p = 0.830 | 1.55 p = 0.655 |
| | (-10.49,11.28) | (-208.69,1787.46) | (3.75,52.22) | (-0.49,0.61) | (-5.42,8.52) |
| WC Δ | -0.13 p = 0.642 | 8.45 p = 0.741 | -0.56 p = 0.370 | 0.01 p = 0.297 | 0.13 p = 0.453 |
| | (-0.69,0.43) | (-42.91,59.80) | (-1.81,0.69) | (-0.01,0.04) | (-0.22,0.49) |
| SBP Δ | 0.10 p = 0.502 | 8.47 p = 0.543 | 0.06 p = 0.860 | -0.01 p = 0.107 | 0.10 p = 0.301 |
| | (-0.20,0.41) | (-19.46,36.40) | (-0.62,0.74) | (-0.03,0.00) | (-0.09,0.30) |
| DBP Δ | 0.02 p = 0.924 | 3.38 p = 0.846 | -0.23 p = 0.589 | 0.01 p = 0.217 | -0.05 p = 0.656 |
| | (-0.36,0.40) | (-31.44,38.19) | (-1.07,0.62) | (-0.01,0.03) | (-0.30,0.19) |

HTN (hypertension); DM (diabetes mellitus); WC (weight change); SBP (systolic blood pressure); DBP (diastolic blood pressure); CRP (c-reactive protein); IgA (immunoglobulin A); IgG (immunoglobulin G); IgM (immunoglobulin M); TGFB (transforming growth factor beta).

those who did not have diabetes, $p<0.05$ (Table 5). Change over the 6 weeks following SG in BMI, systolic blood pressure, and diastolic blood pressure did not correlate significantly with any of the shifts in the markers of inflammation tested (Table 5).

## Discussion

Obesity is a condition of excess adiposity that results in chronic low-grade inflammation. Surgical weight loss procedures produce robust improvements in metabolic indices and increase the quality and longevity of life for the obese individual that is burdened with an array of metabolic comorbidities [25]. In the current study, we asked whether specific inflammatory markers that were previously shown to change after bariatric surgery were altered in BA SG recipients differentially than in WA patients and whether these changes accounted for any variation in change in hip circumference in the six-week time frame following bariatric surgery.

Pre-operatively, BA and WA participants were not significantly different overall; however, there were a few notable differences between the groups. BA participants had a significantly higher hip circumference contributing to a lower waist-to-hip ratio than WA participants. A high waist-to-hip ratio is associated with visceral adiposity whereas a low waist-to-hip ratio is associated with greater subcutaneous fat [26]. Larger proportions of visceral fat lend to an increased risk for cardiovascular disease. Interestingly, WA participants did not have a significant difference in blood pressure or pulse rate in comparison to BA participants, regardless of their larger initial levels of visceral adiposity. This may be due to both groups having a substantial BMI rendering waist-to-hip ratio as a less reliable marker for cardiovascular dysfunction.

Though within the normal range for both groups, there were markedly lower liver enzymes in BA participants compared to liver enzymes of WA participants. ALT and AST are liver enzymes whose elevations are indicative of liver damage. While obesity is associated with an elevation of liver enzymes, black populations tend to have lower liver enzymes when compared to white populations [27]. Additionally, liver enzymes are generally elevated in men in comparison to women [27, 28]. Despite variations in diagnostic tools, blacks have the lowest prevalence of non-alcoholic fatty liver disease (NAFLD), [29] and non-alcoholic steato-hepatitis (NASH) is inversely associated with being African American, though this finding is somewhat limited by non-histologic diagnosis [30]. Further, the temporal severity of advanced fibrosis is elevated in non-Hispanic whites, whereas in non-Hispanic blacks, the trajectory of severity is reduced [31].

Further, a variant of PNPLA3, a pro-steatotic gene that carries with it a higher incidence of NAFLD occurs with the greatest frequency in Hispanics, followed by non-Hispanic whites, and least in African Americans, may explain the lower prevalence of NAFLD in African Americans despite the prevalence of obesity and diabetes in this population [32]. Despite having lower intrahepatic triglyceride accumulation, once NAFLD develops, NASH occurs as frequently and as severe as in Caucasian patients [33].

BA participants enrolled in the present study had a lower hematocrit percentage than WA participants, corresponding to a pattern in the literature showing hematocrit lower in African-Americans than in the white population [34]. Though potentially caused by a variety of factors, anemia is a common contributor to low hematocrit and has a higher prevalence in the black population [35]. Platelet counts were also higher in the BA group compared to WA. This is similar to an earlier study in which black women had significantly higher platelet counts than white women [36], particularly applicable to our study as there were no male BA participants.

As expected, participants had noticeable weight loss in comparison to baseline weight with appreciable improvements in BMI, waist circumference, and hip circumference following SG. A recent JAMA Surgery report suggests that BA do not realize the same positive benefits to excess loss of weight as WA [14]. This report precipitated the present study since the Southeastern U.S. has demographically greater representation of Black Americans, making access to SG surgical participants more likely. Undoubtedly, given the small sample size of the current study, this facet regarding race was not captured in the data set, though %EWL was greater in the WA participants.

Bariatric surgeries are also purported to reduce blood pressure and provide resolution of hypertension along with improvements in heart rate [37, 38], as seen in our cohort. One-third of the U.S. population has high blood pressure and/or is actively treated with anti-hypertensives; this prevalence is known to be substantively higher in the black U.S. community [39]. Though SG uniformly improved systolic blood pressure, diastolic blood pressure remained higher as did pulse rate following SG. It is unknown whether this improvement to cardiovascular health is the result of weight-dependent changes or if the improvement comes from other neural, hormonal, or chemical changes that are weight-independent.

In the current study, IgA and IgM levels were highly variable among participants, both pre-operatively and after six weeks. There was no change to IgA or IgM as a function of surgery in this short time frame. However, the presence of diabetes pre-operatively was linked with lower IgA levels. Poor glycemic control appears associated with an increase in IgA serum antibodies [40]. Within our data set, the patients identified as diabetic may have more controlled glucose, through pharmacologic intervention associated with a reduced IgA in comparison to those individuals who had not been identified to be diabetic. With respect to IgG, BA individuals had higher concentrations at both time points compared to WA, but overall, IgG was dramatically reduced as a result of surgery. IgG is the most common circulating immunoglobulin in

the humoral immune system. It binds to pathogens and protects the body from infection by developing a memory of exposure to specific invaders. Black individuals have previously been assessed to have higher IgG levels in a variety of studies, regardless of context [11, 12]. The reason for the higher IgG levels in black subjects remains unknown. The reason for reduction of IgG levels following surgical weight loss is not currently understood.

IgG-specific antibody mediated reactions are a body's natural defensive reaction to infiltrating food antigens [41]. Following an elimination diet (targeting foods which specifically increased IgG levels), overweight or obese adults were able to decrease IgG antibody titers [42]. Thus, the possibility exists that either as a result of diet choice or genetic factors associated with gastrointestinal permeability, BA have greater levels of IgG that are ameliorated with SG.

CRP is an acute phase, hepatically-derived immune marker for generalized inflammation. As with other markers of inflammation, obesity is associated with higher levels of CRP [43]. Bariatric surgery reduces CRP levels at three months following gastric bypass and as far as twelve months in SG patients [17, 44]. Participants in the current study showed a reduction in CRP levels at six weeks in both BA and WA, although BA individuals had overall higher CRP levels. Interestingly, the change in the reduction of CRP levels at six weeks was comparable in BA and WA when the differential CRP starting levels were controlled for. Overall, this aligns with literature suggesting that black subjects have higher CRP levels when compared to white subjects, and further, that CRP levels are higher in women than in men [45].

TGFβ is an immune cell-derived chemokine which regulates a variety of growth, differentiation, and adhesion cellular processes, in particular the chemotaxis of immune cells. TGFB levels are correlated with obesity in humans [46], and reports show that RYGB decreases TGFβ at one year following surgical weight loss [20]. We show very early reductions in TGFβ in the current work with no differences by race. Circulating levels of TGFβ reductions are not as great as age increased, with older recipients of SG have higher levels of TGFβ. TGFβ reduction after SG is associated with hypertension; this follows accurately with the literature in that TGFβ overproduction is attributed to factors such as elevated angiotensin II, increased blood pressure and increases in sheer stress of fluid dynamics in the body [47]. The role of both age and potentially sex in TGFβ production remains to be further studied.

## Strengths and limitations

The study presented here encompassed patients that were consecutively enrolled in the study. The data collected are representative of short-term weight loss and no information is available about the changes to the immune markers at times greater than six weeks. The trajectories of change in excess weight loss and immune markers may change over time for the population's samples. One of the significant failures of the study is the consent of BA males. Not only do men seek bariatric surgery at reduced rates compared to women, the historical context of BA male utilization in medical studies functions as a barrier to enrollment for this study.

Overall, the sample size is small and given the current trajectories, some parameters may have benefited from increasing the power of the study. While we focused on a limited number of markers, there are vast numbers of others that may have also been interesting to test and are important for future studies. For example, the measurement of adiposity through multifrequency bioelectrical impedance would have garnered, perhaps with better results than the measurement of body weight and BMI.

## Conclusions

In a subset of patients receiving SG representing the demographics of our region of the country, there were no differences in the magnitude of weight loss following SG at six weeks. BA

had greater starting levels of cardiovascular dysfunction, but lower levels of liver dysfunction and reduced tendency for obesity-related lipid disorders. CRP, TGFβ, and IgG were all reduced as a result of SG. IgG was initially elevated in BA in comparison to WA. However, IgG was substantively reduced in BA than WA in the early phase of weight loss. The higher levels of both IgG and CRP in obese BA have significance, given the greater morbidity and mortality of BA individuals in the current viral pandemic [48] where elevations in inflammation appear to exacerbate the severity of symptoms in the BA population [49]. These data suggest that in the early surgical weight loss time frame, markers of immune function are positively improved with SG for both BA and WA but may be improved more so for BA given their higher starting point. Further work is necessary to understand this relationship more adequately.

## Author Contributions

**Conceptualization:** Charles L. Phillips, Bernadette E. Grayson.

**Data curation:** Charles L. Phillips, Bernadette E. Grayson.

**Formal analysis:** Charles L. Phillips, Tran T. Le, Seth T. Lirette, Kenneth D. Vick, Bernadette E. Grayson.

**Funding acquisition:** Bernadette E. Grayson.

**Investigation:** Charles L. Phillips, Bradley A. Welch, Sarah C. Glover, Bernadette E. Grayson.

**Methodology:** Charles L. Phillips.

**Project administration:** Bradley A. Welch, Bernadette E. Grayson.

**Resources:** Bernadette E. Grayson.

**Supervision:** Adam Dungey, Kenneth D. Vick, Bernadette E. Grayson.

**Validation:** Adam Dungey, Kenneth D. Vick, Bernadette E. Grayson.

**Visualization:** Tran T. Le, Seth T. Lirette, Sarah C. Glover, Bernadette E. Grayson.

**Writing – original draft:** Charles L. Phillips, Bernadette E. Grayson.

**Writing – review & editing:** Charles L. Phillips, Tran T. Le, Seth T. Lirette, Bradley A. Welch, Sarah C. Glover, Adam Dungey, Kenneth D. Vick, Bernadette E. Grayson.

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
