## [Decision Letter · Decision Letter 0]

31 Jan 2023

PONE-D-22-23672Immune marker reductions in black and white Americans following sleeve gastrectomy in the short term phase of surgical weight lossPLOS ONE

Dear Dr. Grayson,

Thank you for submitting your manuscript to PLOS ONE. After careful consideration, we feel that it has merit but does not fully meet PLOS ONE’s publication criteria as it currently stands. Therefore, we invite you to submit a revised version of the manuscript that addresses the points raised during the review process.

Major concern with the statistical methods which are not explained well, Kindly, elaborate statistical methods and highlight the importance of the studies. 

We look forward to receiving your revised manuscript.

Kind regards,

Rasheed Ahmad, Ph.D.

Academic Editor

PLOS ONE

Journal Requirements:

2. Please include your tables as part of your main manuscript and remove the individual files. Please note that supplementary tables (should remain/ be uploaded) as separate "supporting information" files"

Reviewers' comments:

Reviewer's Responses to Questions

**Comments to the Author**

1. Is the manuscript technically sound, and do the data support the conclusions?

Reviewer #1: Partly

Reviewer #2: Partly

2. Has the statistical analysis been performed appropriately and rigorously? 

Reviewer #1: Yes

Reviewer #2: No

3. Have the authors made all data underlying the findings in their manuscript fully available?

Reviewer #1: Yes

Reviewer #2: Yes

4. Is the manuscript presented in an intelligible fashion and written in standard English?

Reviewer #1: No

Reviewer #2: Yes

5. Review Comments to the Author

Reviewer #1: The authors of the paper determined if IgG, IgM, IgA, TGFβ, and CRP change similarly in black and white Americans after sleeve gastrectomy.

While the rationale of the study seems justified, there are some important problems in the article. Aim, hypothesis, study questions, statistical methods to test these questions, and a logical presentation of the results and discussion unfortunately all seem weak.

What was the reason for selecting IgG, IgM, IgA, TGFβ, and CRP, specifically?

Introduction: First paragraph. References are missing.

The statistical methods are unclear, and it is difficult to follow what is the study question for which each of the tests is performed.

For example:

‘In addition to the adjusters listed below’

Please clarify what are these adjusters? Below is mentioned the adjusters for change, as well as for ‘all models were adjusted for…’

‘Weight change was modeled using ordinary least squares on the difference in weight post-op vs. pre-op. Similar models were used to model immune marker change, additionally adjusting for BMI change, systolic change, and diastolic change.’

It is unclear where these models were used (what is the study question, predictor, outcome).

The models are not adjusted for multiple testing. For example, in Table 5, none of the p values would likely remain significant if multiple comparisons were taken into account.

The results section is inconsistent.

There seems to be no real logic and in every paragraph, the topics jump from one area to another.

For example, the first paragraph of the results mixes sex, age and BA/Wa comparison.

Row 149-

The percentages are unclear regarding total, f/m.

BA had no males but 6/34 of WA were males. Was there a difference in immune markers or clinical conditions between the sexes in WA?

The next parag (row 157) continues with ‘Pre-operative Characteristics of BA and WA Participants’, but wasn’t there already preop comparisons in the previous parag?

The inconsistency continues in the next parag. The text jumps from influence of clinical characteristics to comparisons of BA/WA and back to clinical characteristics.

In the text, what test is performed for ‘influence’ or ‘association’. The text also refers to ‘correlation’. What is that test?

The immune marker results are drown to the quite expected weight + clinical characteristic text.

‘Plasma IgG varied significantly by race with BA having higher levels of IgG than WA, p<0.001.’ What time point is this referring to?

‘Furthermore, BA had greater reductions in IgG, with an average reduction of 1108 mg/dL as a result of SG in comparison to WA (TABLE 5).’

Is the %change in IgG between BA/WA different. Usually, reductions are greater if the baseline level is higher.

Please use exact p values instead of p<0.05 etc.

Discussion

The discussion lacks focus. For example, there are several paragraphs of ALT, NAFLD, hematocrit, platelet count, although these were not mentioned in the results text and are not relevant for the study question.

The discussion also presents new data that is not in the results, like:

‘Participants in the current study showed a reduction in CRP levels at six weeks in both BA and WA, although BA individuals had overall higher CRP levels.’

CRP was not mentioned in the results.

After the discussion, the biological or clinical conclusion of the study remains unclear.

Table 1, what does Y mean?

What do the percentages mean?

In Figure 1: What is p(subjects). What time point is meant for p(race)?

Figure 1 appears between the tables

Reviewer #2: It is recommended to emphasize that bariatric surgery, due to professional ethics, is a treatment for obesity but not first-line, and it is not recommended for patients who have not undergone the first-choice treatments: nutrition and pharmacology, or in those to whom it is necessary to avoid a mortal risk.

6. PLOS authors have the option to publish the peer review history of their article (what does this mean?). If published, this will include your full peer review and any attached files.

Reviewer #1: No

Reviewer #2: **Yes: **Melchor Alpízar-Salazar

---

## [Author Response · Author response to Decision Letter 0]

11 May 2023

PONE-D-22-23672

5.10.23

Please see the updated comment for the Funding and also we changed the name of track changes file. It was all there highlighted. Please look carefully to find it. We ask for expedited re-review of this because of a grant deadline. The PLOSONE emails are going to our quarantine page and take days to identify. 

Thank you

B

The authors of the paper determined if IgG, IgM, IgA, TGFβ, and CRP change similarly in black and white Americans after sleeve gastrectomy.

While the rationale of the study seems justified, there are some important problems in the article. Aim, hypothesis, study questions, statistical methods to test these questions, and a logical presentation of the results and discussion unfortunately all seem weak. 

What was the reason for selecting IgG, IgM, IgA, TGFβ, and CRP, specifically?

RESPONSE: As demonstrated in paragraph three of the Introduction, BA carry a high burden of Inflammation and as demonstrated in paragraph four, IgG, IgM, IgA, TGFβ, and CRP had been previously reported to be changed In other populations. Further, IgG has been previously shown to be different In BA population.

Introduction: First paragraph. References are missing.

RESPONSE: References have been added.

The statistical methods are unclear, and it is difficult to follow what is the study question for which each of the tests is performed. 

For example:

‘In addition to the adjusters listed below’

Please clarify what are these adjusters? Below is mentioned the adjusters for change, as well as for ‘all models were adjusted for…’

 ‘Weight change was modeled using ordinary least squares on the difference in weight post-op vs. pre-op. Similar models were used to model immune marker change, additionally adjusting for BMI change, systolic change, and diastolic change.’

It is unclear where these models were used (what is the study question, predictor, outcome).

RESPONSE: We have updated the Statistical Analysis section of the manuscript to more clearly describe the analyses done. We thank the reviewer for the opportunity to make these clarifications.

The models are not adjusted for multiple testing. For example, in Table 5, none of the p values would likely remain significant if multiple comparisons were taken into account.

Response: Since this is a hypothesis generating manuscript, as opposed to a hypothesis confirming one, multiple comparisons aren’t of a concern, statistically (ref). While we agree that most of the p-values don’t convey strong results, we would prefer to leave these as-is to assist the reader in making comparisons across the many models that we performed.

Andrew Gelman, Jennifer Hill & Masanao Yajima (2012) Why We (Usually) Don't Have to Worry About Multiple Comparisons, Journal of Research on Educational Effectiveness, 5:2, 189-211, DOI: 10.1080/19345747.2011.618213

The results section is inconsistent.

There seems to be no real logic and in every paragraph, the topics jump from one area to another. 

For example, the first paragraph of the results mixes sex, age and BA/Wa comparison.

Row 149-

The percentages are unclear regarding total, f/m.

BA had no males but 6/34 of WA were males. Was there a difference in immune markers or clinical conditions between the sexes in WA?

The next parag (row 157) continues with ‘Pre-operative Characteristics of BA and WA Participants’, but wasn’t there already preop comparisons in the previous parag?

The inconsistency continues in the next parag. The text jumps from influence of clinical characteristics to comparisons of BA/WA and back to clinical characteristics. 

RESPONSE: We were moving first through PRE-data and then addressing POST data. But now, we have reformatted and we narrate through each Table and Figure in order. 

In the text, what test is performed for ‘influence’ or ‘association’. The text also refers to ‘correlation’. What is that test?

RESPONSE: We apologize for this wording. We have removed influence and correlation. The logistic regression modelling establishes relationships between factors and we feel that “associate” is a reasonable word to use for this. We refer the reviewer to the updated Statistical Analysis section which should now provide the answers to these questions. 

The immune marker results are drown to the quite expected weight + clinical characteristic text.

RESPONSE: We are not sure what this comment means.

‘Plasma IgG varied significantly by race with BA having higher levels of IgG than WA, p<0.001.’ What time point is this referring to?

RESPONSE: This is a main effect of time as well as Student’s t test at pre-op and 6 weeks. 

‘Furthermore, BA had greater reductions in IgG, with an average reduction of 1108 mg/dL as a result of SG in comparison to WA (TABLE 5).’

Is the %change in IgG between BA/WA different. Usually, reductions are greater if the baseline level is higher.

RESPONSE: As the reviewer suggested, TABLE 3 does show that pre-op levels of IgG were, on average higher in BA than WA. And we agree with the point the reviewer raise about floor effects possibly lower the range that WA IgG values can reduce. To that end, we have added text to the manuscript to address this. 

Please use exact p values instead of p<0.05 etc.

This has been included in the tables. 

Discussion

The discussion lacks focus. For example, there are several paragraphs of ALT, NAFLD, hematocrit, platelet count, although these were not mentioned in the results text and are not relevant for the study question.

The discussion also presents new data that is not in the results, like:

‘Participants in the current study showed a reduction in CRP levels at six weeks in both BA and WA, although BA individuals had overall higher CRP levels.’

CRP was not mentioned in the results. 

RESPONSE: IT is now mentioned in the results. 

After the discussion, the biological or clinical conclusion of the study remains unclear.

Table 1, what does Y mean?

RESPONSE: Y means yes

What do the percentages mean?

RESPONSE: Percentages mean the Percentage of Total

In Figure 1: What is p(subjects). What time point is meant for p(race)?

RESPONSE: We have clarified this.

Figure 1 appears between the tables 

RESPONSE: This is how we organize the data.

---

## [Editor Report · Decision Letter 1]

6 Jul 2023

Immune marker reductions in black and white Americans following sleeve gastrectomy in the short term phase of surgical weight loss

PONE-D-22-23672R1

Dear Dr. Grayson,

We’re pleased to inform you that your manuscript has been judged scientifically suitable for publication and will be formally accepted for publication once it meets all outstanding technical requirements.

Kind regards,

Rasheed Ahmad, Ph.D.

Academic Editor

PLOS ONE
---

## [Editor Report · Acceptance letter]

13 Jul 2023

PONE-D-22-23672R1 

Immune marker reductions in black and white Americans following sleeve gastrectomy in the short-term phase of surgical weight loss 

Dear Dr. Grayson:

I'm pleased to inform you that your manuscript has been deemed suitable for publication in PLOS ONE. Congratulations! Your manuscript is now with our production department. 

Kind regards, 

on behalf of

Dr. Rasheed Ahmad 

Academic Editor

PLOS ONE